# Investigation of Antioxidant and Cytotoxicity Activities of Chocolate Fortified with Muscadine Grape Pomace

**DOI:** 10.3390/foods12173153

**Published:** 2023-08-22

**Authors:** Ahmed G. Darwish, Islam El-Sharkawy, Chunya Tang, Qinchun Rao, Juzhong Tan

**Affiliations:** 1Center for Viticulture and Small Fruit Research, College of Agriculture and Food Sciences, Florida A&M University, Tallahassee, FL 32308, USA; ahmed.darwish@famu.edu (A.G.D.); islam.elsharkawy@famu.edu (I.E.-S.); 2Department of Biochemistry, Faculty of Agriculture, Minia University, Minia 61519, Egypt; 3Department of Nutrition and Integrative Physiology, Florida State University, Tallahassee, FL 32306, USA; ct19g@fsu.edu; 4Department of Animal and Food Science, University of Delaware, Newark, DE 19716, USA

**Keywords:** cytotoxicity, DPPH, FRAP, MDA-MB-468, phenolics, flavonoid

## Abstract

Muscadine grape pomace and mixed products with chocolate extracts from three muscadine genotypes exhibiting different berry skin colors (black and bronze) were investigated for total phenolic content (TPC), total flavonoid content (TFC), DPPH, FRAP antioxidant activity, and anticancer activity using MDA-MB-468 (MM-468; African American) breast cancer cells. Muscadine berry extracts and mixed products showed cytotoxicity activities of up to 70% against MM-468 breast cancer cells. Cell growth inhibition was higher in ‘macerated Floriana’ with an IC50 value of 20.70 ± 2.43 followed by ‘Alachua’ with an IC50 value of 22.25 ± 2.47. TPC and TFC in macerated MGP powder were (1.4 ± 0.14 and 0.45 ± 0.01 GAE/g FW, respectively), which was significantly higher than those in cocoa powder. Data analysis showed a high association between DPPH, FRAP antioxidant activities, and TPC content and a positive high correlation between anticancer activity and antioxidant capacity and between TPC and anticancer activity. The anticancer and antioxidant effects of muscadine grape pomace and chocolate extracts are attributed to the TPC of extracts, which showed a stronger positive correlation with growth inhibition of African American breast cancer cells. This study would be of great value for food industries as well as other manufacturers who are interested in new food blends.

## 1. Introduction

Muscadine grapes (*Vitis rotundifolia*) are widely grown in the southeastern U.S., including the state of Florida. Muscadine grape pomace (MGP), which consists of skin and seed, is a winery or juice-processing waste product and makes up approximately 20–25% of the grape weight [1]. Previous studies have reported that MGP has diverse and high concentrations of polyphenolic compounds such as phenolic acid, procyanidins, and anthocyanins [1,2,3]. Those bioactive constituents from MGP have been reported to possess potent antioxidant, anticancer, antimutagenic, and anti-inflammatory properties in multiple previous studies [2,4,5,6]. In addition, MGP is a good source of dietary fiber, and contains more than 60% dietary fiber by dry weight [7]. Dietary fibers have been identified as good nutrient fortifiers and fat replacers for many food products such as chocolates [8,9,10]. Although MGP is a potential value-added ingredient for food products, it is underused and discarded after juice production or winemaking. The global wine industry produces approximately 14 million tons of pomace every year and most of the pomace is landfilled [11]. Direct disposal of grape pomace in landfill can cause many negative environmental effects including contaminating surface and groundwater, attracting agricultural pests and flies which may spread plant and human diseases, and generating odor [11,12,13].

Chocolate is a semi-solid suspension of fine solid particles of sugar, cocoa, and, possibly, milk solids in a continuous phase of cocoa butter. The United States is the leading manufacturer and market of chocolate. It accounts for approximately 30% of total world chocolate production and creates around 20 billion retail sales per year [14]. Commercially available chocolates contain 0.9–4.0% of polyphenols and <7% of dietary fiber by dry weight. On the other hand, MGP contains more than 12% polyphenols and >60% dietary fiber by dry weight [15]; therefore, infusion of MGP into chocolate as a substitute for cocoa solid can potentially reduce the high-fat content in the diet and improve adipokine secretion and oxidative stress [16]. Also, food supplemented with MGP can act on insulin-sensitive cells, affect the function of pancreatic β-cells, and prevent the effect of a high-fat diet on pancreatic insulin secretion and lipid accumulation, thus improving insulin resistance without downgrading the texture and enhancing the antioxidant and anticancer activities of the chocolates.

Many previous studies have investigated the antimicrobial [17,18], antioxidant [1,19,20], and anticancer activities [21,22,23] of MGP. The antioxidant and anticancer activities of cocoa and chocolate products, especially dark chocolates, have also been intensively studied [24,25,26,27,28]. However, no previous studies have reported the production of chocolate using MGP as one value-added ingredient. The sensory attributes, mechanical properties, and digestibility of chocolates enriched with grape pomaces were reported in some previous studies [29,30]. However, the antioxidant and anticancer activities of grape pomace-enriched chocolates, especially MGP-enriched chocolates, were barely investigated. The variety and the processing of MGP can significantly influence the bioactive compounds, such as polyphenolic compounds, in the MGP. Previous studies evaluated the polyphenolic content and antioxidant activity of different muscadine grape varieties and genotypes in the United States [6,31], and some MGP varieties have almost three times the total phenolic content and antioxidant activity of other MGP varieties. The anticancer activity of MGP has been examined using different cancer cell lines and showed significantly different cytotoxicity among different varieties [21,32].

The preparation and processing of MGP and MGP chocolates can potentially influence antioxidant and anticancer activities. Muscadine grape pomace may be obtained either after the maceration of winemaking or after pressing for juice. Maceration refers to soaking the grape juice (before or during fermentation) along with the crushed skins, seeds, and stalks (pomace). It is a required process for red wine, where the bioactive compounds such as tannins, coloring agents (anthocyanins), and flavor compounds, leach from the pomace into the must. Previous studies have indicated that the maceration process can change the profile of bioactive compounds in grape pomace, and therefore influence the antioxidant activity [33,34]. The processing, especially when it involves lengthy thermal treatment and/or high shear, can also impact the antioxidant and anticancer activities of chocolates, because using high shear and thermal treatment during processing may have negative effects on the polyphenols which are considered the main source of antioxidants and anticancer agents [35]. Chocolates typically required a lengthy conching process, which subjects the ingredients to repeated crushing or shear and relatively high temperature (49–82 °C) for 12–96 h, to reduce particle size and develop flavors [36]. The conching process can result in the loss of bioactive compounds and therefore reduce the antioxidant and anticancer activities of the chocolates due to extreme processing [37,38].

Due to the knowledge gap in the antioxidant and anticancer activities of MGP-enriched chocolates, this study was designed to investigate the effects of MGP concentration, variety, and processing on the antioxidant and anticancer activities of MGP chocolates. In the study, three varieties of MGP were obtained before and after maceration and were used as ingredients to produce chocolates by partially substituting for the cocoa solids. The total phenolic content in the MGP chocolates was measured, and the antioxidant and anticancer activities were evaluated by in vitro models.

## 2. Materials and Methods

The schematic diagram (Figure 1) below shows the methodologies for MGP chocolate preparation, and the analysis of its qualities.

### 2.1. Preparation of MGP Powder and MGP Chocolates

MGPs from three varieties (Floriana, Senoria, and Alachua) of muscadine grapes were collected from the vineyards at Florida A&M University (FAMU) Center for Viticulture and Small Fruit Research. The MGPs contained grape skins, seeds, and stalks. The Floriana MGP was obtained before and after maceration, while the Senoria and Alachua MGPs were collected without going through maceration. The MGPs were freeze-dried in a Harvest Right commercial freeze dyer at vacuum <500 mTorr and ~−40 °C for 72 h. The dried MGPs were ground into fine powders by a Nutrimill stone grain mill and passed through a stainless steel 200-mesh sieve. The MGP powder was vacuum-packed in an aluminum pouch and stored in a −20 °C freezer before use. Defatted raw cocoa powder (Navitas Organics, Novato, CA, USA) and raw cocoa butter (Terrasoul Superfoods, Fort Worth, TX, USA) were purchased from Amazon.

Each MGP powder sample was mixed with cocoa powder into a mixture that was 21% of the total ingredient weight. The concentration of MGP powder was 5%, 7%, or 10% of the total ingredient weight. Our preliminary studies had shown that at an MGP concentration of less than 10%, the color and sensory attributes (sourness and overall likeness) of the MGP chocolate were not significantly different from the control sample. Therefore, varied MGP concentrations below 10% by total ingredient weight were applied to the MGP chocolate samples. The cocoa−MGP mixture was refined/conched with cocoa butter (30% of total ingredient weight) and granulated sugar (49% of total ingredient weight) in a melanger for 24 h to make chocolate liquid. The chocolate liquid was tempered by using a KitchenAid Precise Heat Mixing Bowl, and tempered chocolate liquid was poured into chocolate molds and held at 20 °C to solidify. One control sample, which contained 21% (*w*/*w*) cocoa powder, 30% (*w*/*w*) cocoa butter, and 49% (*w*/*w*) sugar, was produced following the same protocol mentioned above.

### 2.2. Chemicals

The chemicals used in this study, including Folin–Ciocalteu phenol reagent, 2,2-diphenyl-1-picrylhydrazyl (DPPH), gallic acid, quercetin, Trolox, HPLC-grade methanol, 2,4,6-tripyridyl-s-triazine (TPTZ), glacial acetic acid, sodium acetate trihydrate, and ferric chloride (FeCl_3_), were purchased from Sigma (Sigma-Aldrich, St. Louis, MO, USA).

### 2.3. Preparation of Cocoa Beans and Chocolate Extracts

All roasted MGP power and chocolate samples were ground to a fine powder using a Geno/Grinder 2010 (Metuchen, NJ, USA). Next, eight grams of sample powder were subjected to methanol extraction using 100 mL of methanol. All extractions were performed under shaking (150 rpm) for 24 h in the dark at room temperature. Then, the extracts were filtered through Whatman No. 41 filter papers (Thomas Scientific, Swedesboro, NJ, USA). The collected supernatant was concentrated using a Heidolph rotary evaporator (Thermo Fisher Scientific, Waltham, MA, USA) at 40 °C and then dehydrated using a speed vacuum (Eppendorf, Enfield, CT, USA). All dried extracts were stored at 4 °C in the dark for further analysis. The stock solution of cocoa extracts was prepared at a 10 mg/mL concentration in DMSO to determine the total metabolite content and antioxidant activity.

### 2.4. Analysis of Total Phenolic and Flavonoid Contents

Total phenolic content (TPC) was measured according to the Folin–Ciocalteu colorimetric method based on the previously described protocol with minor modifications [6]. Briefly, 15 µL of diluted samples was placed in a 96-well microplate (Genesee Scientific, San Diego, CA, USA). Subsequently, 240 µL of Folin–Ciocalteu reagent (1:15, *v*/*v*) was added to the wells, mixed with samples, and incubated in the dark at room temperature for 30 min. Then, the mixtures were treated with 15 µL of 20% sodium carbonate solution. The mixture was shaken before measuring the absorbance at λ = 755 nm using a microplate reader (ACCURIS SmartReader; Edison, NJ, USA). Gallic acid solutions in the specific concentration range were used to construct a calibration curve. TPC estimation was performed in triplicate (*n* = 3) and expressed as milligram gallic acid equivalents per gram of sample fresh weight (mg GAE/g FW).

Total flavonoid content (TFC) was measured based on the previously reported method with some modifications [39]. Briefly, an aliquot (25 µL) of diluted samples was mixed with 75 µL of 96% methanol (*v*/*v*) and placed in a 96-well microplate. Then, 5 µL of 10% aluminum chloride and 5 µL of potassium acetate (1M) were added to the mixture. Finally, 140 µL of distilled water was added and the mixture was incubated for 30 min in the dark at room temperature. The mixture was shaken before measuring the absorbance at λ = 415 nm using a microplate reader. Quercetin was used to construct the calibration curve in a different concentration range. Total flavonoid content was estimated in triplicate (*n* = 3) and expressed as milligram quercetin equivalents per gram of sample fresh weight (mg QE/g FW).

### 2.5. DPPH Radical-Scavenging Activity

DPPH radical-scavenging activity was evaluated based on the protocol developed in a previous study [40]. Briefly, 100 µL of diluted sample (freeze-dried MGP, MGP chocolates, or cocoa powder) was mixed with 100 µL of freshly prepared DPPH methanolic solution (200 µM). The mixture was then incubated for 30 min in the dark at room temperature. The absorbance was measured at λ = 515 nm using a microplate reader. As a control, DMSO was used in place of the samples. The DPPH-scavenging activity was measured in triplicate (*n* = 3). The data were expressed as the percentage scavenging of DPPH radicals and calculated using the following equation:DPPH (%) = [1 − (A_sample_ − A_background_)/(A_DMSO_ − A_background_)] × 100

### 2.6. Ferric Reducing Antioxidant Potential (FRAP) Assay

The FRAP assay was conducted based on the methods developed in a previous study [41]. FRAP reagent was prepared at the ratio of 10:1:1 (*v*/*v*/*v*) comprising 300 mM acetate buffer (pH 3.6), a solution of 40 mM TPTZ in 40 mM HCl, and 20 mM FeCl3. Freshly prepared FRAP reagent (280 µL) and diluted samples (20 µL) were mixed in a 96-well microplate and incubated at 37 °C in the dark for 30 min. The absorbance was measured at λ = 590 nm using the microplate reader. Trolox at different concentrations was used to make the standard curve. FRAP radical-scavenging activity was estimated in triplicate (*n* = 3). Data are expressed in micro-molar Trolox equivalents per gram of sample fresh weight (µM TE/g FW).

### 2.7. Cell Culture

Triple-negative breast cancer cells MDA-MB-468 (MM-468) (African American) were purchased from the American Type Culture Collection (ATCC) (Manassas, VA, USA) and used to evaluate the anticancer activities of MGP and MGP chocolate samples. The cells were grown in 75 mm flasks using DMEM, 10% heat-inactivated fetal bovine serum (FBS), and 1% penicillin/streptomycin (100 U/mL penicillin and 0.1 mg/mL streptomycin). Cell cultures were incubated in an incubator with a controlled atmosphere of 5% CO_2_ and 37 °C temperature.

### 2.8. Anticancer Activity

Cells (density of 5 × 10^3^ cells/well) were incubated overnight in experimental media in 96-well plates. All extracts were prepared in 3 biological replicates, and each replicate was tested in triplicate on the 96-well plate [21]. Cocoa extracts (dissolved in DMSO) were added to the 96-well plates at final concentrations of 1 mg/reaction (10 μg/μL). Cells were incubated at 37 °C for 72 h, and 100 μL of MTT solution (5.5 mg/mL) was pipetted into the plate and incubated for an additional 1.5 h. The fluorescence signal was measured (540/580 nm) using a microplate reader (ACCURIS Smart Reader; Edison, NJ, USA). Controls were treated with DMSO at the same concentration as that used in the extracts (<1%). Blank wells contained only media, without cells. The cytotoxicity rate was calculated based on cell viability, using the following formula:% Inhibition = [1 − (A_sample_ − A_blank_)/(A_control_ − A_blank_)] × 100

The IC50 for cell growth inhibition was calculated for extracts exhibiting the highest activity. The IC50 was calculated using the online tool AAT Bioquest IC50 Caculator (https://www.aatbio.com/tools/ic50-calculator, accessed on 20 April 2023).

### 2.9. Data Analysis

All measurements, including the antioxidant activities, anticancer activities, TFC contents, and TPC contents, of the samples (freeze-dried MGP, MGP chocolate, and cocoa powder) were triplicated. The data were expressed as the mean and standard deviation (mean ± standard deviation). The significance of differences between different treatment groups was evaluated statistically by one-way factorial analyses of variance (ANOVA) and post hoc multiple comparisons using SAS 9.4. The significant level (*p*-value) of the ANOVA tests was set at 0.05.

## 3. Results and Discussion

### 3.1. Total Phenolic Content (TPC) and Total Flavonoid Content (TFC) in MGP Powder and MGP Chocolates

The TPC and TFC in MGP powder and cocoa powder samples is shown in Figure 2. The TPC and TFC in unmacerated MGP powder ranged from 0.07 ± 0.03 to 0.2 ± 0.03 mg GAE/g FW and 0.11 ± 0.00 to 0.13 ± 0.00 mg GAE/g FW, respectively, which was significantly lower than in cocoa powder (0.86 ± 0.1 GAE/g FW TPC and 0.24 ± 0.01 GAE/g FW TFC). This observation concurred with many previous studies, which reported that cocoa has approximately one to five times more TPC and TFC than wine, black tea, green tea [42,43], and MGP extract [1]. However, the TPC and TFC in macerated MGP powder was 1.4 ± 0.14 GAE/g FW and 0.45 ± 0.01 GAE/g FW, which was significantly higher than in cocoa powder. Previous studies [44,45] have shown that the TPC and TFC in grape pomace decreased after maceration. The disparity is because the previous studies reported the TPC and TFC in macerated grape pomace without drying, however, the macerated MGP samples were freeze-dried before measuring the TPC and TFC. During maceration, on the one hand, the TPC and TFC in MGP reduced due to the phenolic and flavonoid compounds leaching into the grape juice; on the other hand, the yeast and other microorganism transferred carbohydrates (e.g., sugar) and nitrogenous compounds into volatile compounds, such as alcohols and acetaldehyde [46]. A significant quantity of volatiles was removed during the drying process, which increased the concentration of TFC and TPC in macerated MGP.

The TPC and TFC in MGP chocolates is shown in Table 1. Generally, the TPC and TFC in unmacerated Senoria, Floriana, and Alachua MGP chocolates were higher than the those of their MGP powders because cocoa powder, which contains high TPC and TFC, was formulated into the chocolates. However, the TPC and TFC in the control sample and macerated Floriana MGP chocolates were lower than the ones of cocoa powder and macerated Floriana MGP powder. This was due to there being no phenolic-containing and no flavonoid-containing ingredients and no sugar and cocoa butter, which were added to the chocolates. In addition, the conching processing of the chocolates reduced the amount of TPC and TFC due to long exposure to high temperatures [47,48]. With the increased concentration of macerated Floriana MGP, the TPC and TFC in the chocolates were higher. However, with an increased concentration of unmacerated MGP, the TPC and TFC in the chocolates were lower. These observations concur with the data in Figure 2, indicating that unmacerated MGP has less TPC and TFC than cocoa powder.

### 3.2. The Antioxidant Activities of MGP Chocolates

The antioxidant activities of MGP and cocoa powder (control) are shown in Figure 3. Generally, the unmacerated MGP had similar antioxidant activities with both DPPH and FRAP approaches. The cocoa powder had a stronger antioxidant activity than all unmacerated MGP, however, macerated MGP had a comparable or higher antioxidant activity than that of cocoa powder. Phenolic and flavonoid compounds can deactivate free radicals based on their ability to donate hydrogen atoms to free radicals. Previous studies [49,50] have indicated that TFC and TPC of foods have strong correlations with antioxidant activities, and higher TFC and TPC in food materials typically result in stronger antioxidant activities. The antioxidant activities of MGP shown in Figure 3 were dependent on the TFC and TPC of MGP shown in Figure 2. The antioxidant activities of freeze-dried MGP measured by DPPH and FRAP approaches generally agreed with each other, with one exception. The FRAP approach indicated that the antioxidant activity of macerated MGP was significantly higher than that of all the samples; however, the DPPH approach indicated that macerated MGP had a similar antioxidant activity to the control sample. This is due to several phenolic compounds being extracted from the MGP extracts employed during the maceration process and this enrichment implies a higher antioxidant activity of the macerated MGP which is related to the phenolic compounds found in the macerated MGP, which include those with higher antioxidant activity such as malvidin, cyanidin, catechin and caffeic acid, and cinnamic and gallic acids [51].

The antioxidant activities of MGP chocolates based on DPPH and FRAP approaches are shown in Table 2. Overall, the measurements by DPPH and FRAP approaches agreed with each other. The macerated MGP chocolate samples had a significantly higher antioxidant activity than all the unmacerated MGP chocolate samples and the control sample. With the increased concentration of macerated MGP, the antioxidant activities of the chocolates were higher. This was due to higher TFC and TPC in macerated MGP formulated into the chocolate samples. Increasing unmacerated MGP concentrations did not necessarily reduce or increase the antioxidant activities of the chocolates since TPC and TFC are not the only bioactive compounds contributing to the antioxidant activities. Increased concentrations of unmacerated MGP in the chocolates reduced the TPC and TFC in the chocolate samples; however, many previous studies [52,53] have reported that non-phenolic bioactive compounds such as ascorbic acid, also contribute to the antioxidant activities of grapes.

### 3.3. The Anticancer Activities of MGP Chocolates

The potential cancer activities of MGP extracts were investigated based on their nutritional benefits. MGPs from three varieties (Floriana, Senoria, and Alachua) of muscadine grapes were examined on triple-negative breast cancer cells MDA-MB-468 (MM-468) (African American) at the dose of 12.5 μg/mL, 25 μg/mL, and 50 μg/mL. The cytotoxicities (%) of the cancer cells that were exposed to MGP extracts were shown in Figure 4. Our results showed that macerated Floriana had higher anticancer activity. The acquired IC50 values indicated that the extract concentration to achieve 50% cell growth inhibition was higher in macerated Floriana with an IC50 value of 20.70 ± 2.43 followed by Alachua with IC50 22.25 ± 2.47. However, Senoria MGP and Floriana MGP have not shown significant anticancer activities compared with cocoa powder. This data suggested a high association between TPC, antioxidant, and anticancer activities.

The anticancer activities of the control chocolate sample and selected MGP chocolate samples (Alachua MGP chocolates) against triple-negative breast cancer cells are shown in Figure 5. The control sample had shown some anticancer activity, however, the anticancer activity is not comparable to the anticancer activity of cocoa powder. The reduction in anticancer activity in the control chocolate sample was mainly due to the formulation of ingredients without anticancer activity including sugar and cocoa butter.

Although all the MGPs showed anticancer activities against breast cancer cells, not all MGP chocolate samples showed anticancer activity. Namely, only Alachua MGP chocolate samples showed anticancer activities against breast cancer cells, while Floriana and Senoria MGP chocolate samples did not show any anticancer activities, resulting in 0% cytotoxicity for triple-negative breast cancer cells. This was due to the ingredients (MGP, cocoa powder, sugar, and butter) of the MGP chocolate samples being subjected to heat and shear force in the conching process for a relatively long period (~50 °C for 24 h). A long exposure to heat and shear resulted in the reduction of particle size, breakage of muscadine grape cells, and reduction in anticancer-related bioactive compounds [54,55,56]. Alachua MGP chocolate samples were one exception, and had high anticancer activity against breast cancer cells, and increased Alachua MGP content in the chocolate sample increased anticancer activity. One possible explanation for this observation is that MGP from Alachua grapes is more resistant to shear than MGP from other grape varieties, resulting in a slower release of bioactive compounds from the grape cells. Therefore, the anticancer-related bioactive compounds in Alachua MGP remained relatively intact throughout the conching process.

## 4. Conclusions

In this study, we investigated the variations in total phenolics, flavonoids, antioxidant activity, and cytotoxicity among muscadine grape pomace and mixed products with chocolate extracts from three muscadine genotypes exhibiting different berry skin colors (black and bronze). Our results collectively indicated a differential effect of muscadine grape pomace and chocolate product extracts in cytotoxicity activity using African American breast cancer (MM-468) cell lines. In addition, there was a high association between cytotoxicity and antioxidant activities, and total phenolic content. Among the different muscadine pomace and chocolate product extracts tested in this study macerated Floriana and Alachua showed the highest cytotoxicity activity compared with other extracts. Our results also confirmed that the extract which contained higher phenolic and flavonoid levels as well as a higher antioxidant showed higher cytotoxicity activity than the ones with lower phenolic and flavonoid contents. Overall, the present study provides a framework to understand the enhancement of chocolate’s nutritional value by fortifying it with muscadine grape pomace (MGP). In addition, we investigated the variations in total phenolic and flavonoid contents, antioxidants, and cytotoxicity activities for the fortified products. Cytotoxicity activity is anticipated to be of great value to several entities in the food and nutraceutical sectors.

## Figures and Tables

**Figure 1 foods-12-03153-f001:**
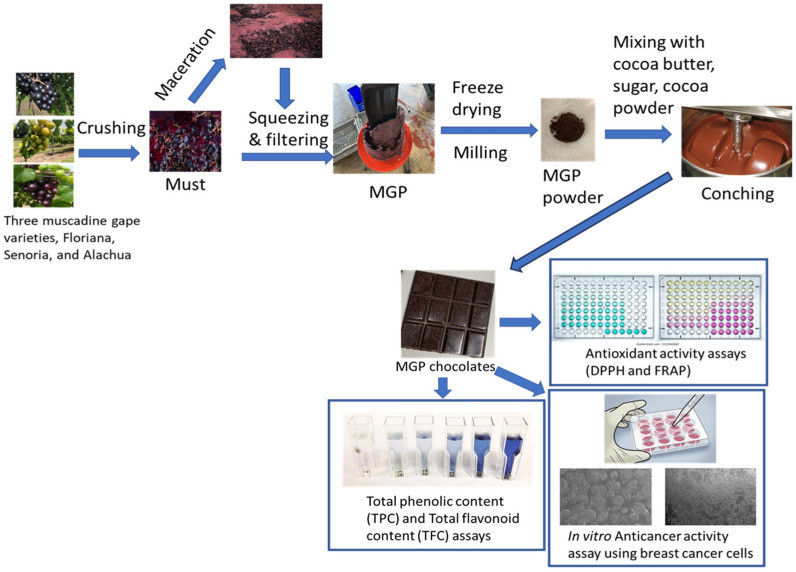
Schematic of the preparation of MGP chocolates and quality analysis.

**Figure 2 foods-12-03153-f002:**
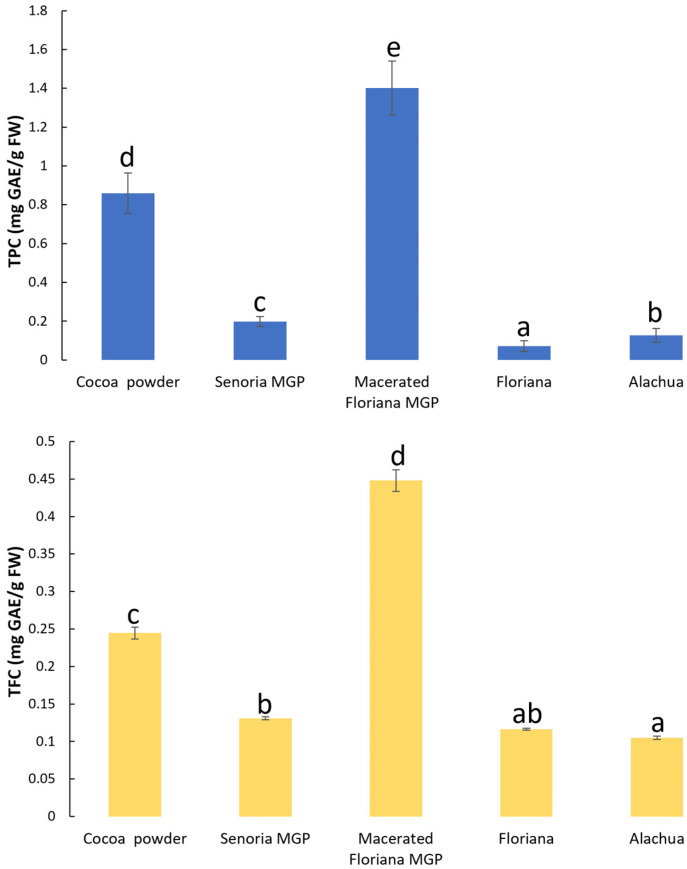
The total phenolic content (TPC) and total flavonoid content (TFC) in cocoa power and freeze-dried MGP powder of three muscadine varieties (Floriana, Senoria, and Alachua) and one macerated MGP dry powder (Floriana). Different letters indicate significant difference at *p*-value < 0.05.

**Figure 3 foods-12-03153-f003:**
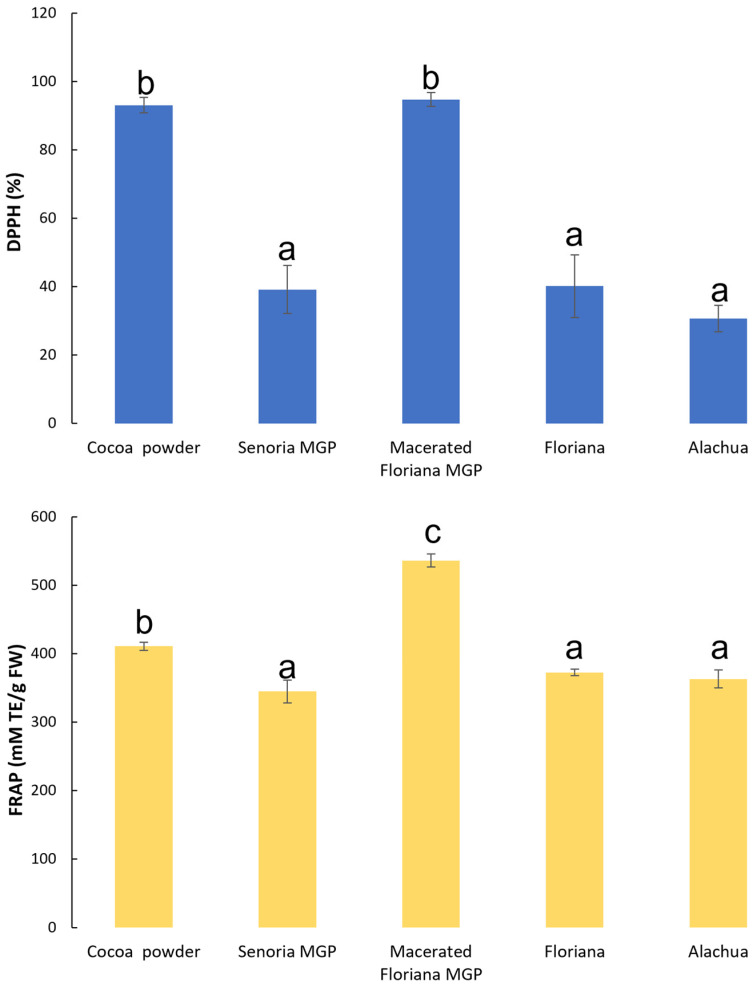
The antioxidant activities of freeze-dried MGP samples of three muscadine varieties (Floriana, Senoria, and Alachua) and one macerated MGP (Floriana). The antioxidant activities were measured based on the DPPH and FRAP approaches described in Section 3.2. Different letters indicate significant difference at *p*-value < 0.05.

**Figure 4 foods-12-03153-f004:**
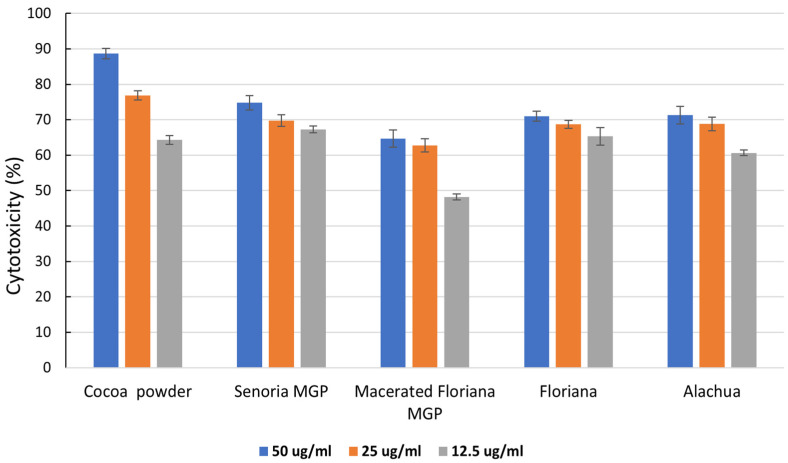
Inhibition percentage of triple-negative breast cancer cells MDA-MB-468 (MM-468) (African American) by MGP extract samples of three muscadine varieties (Floriana, Senoria, and Alachua) and one macerated MGP (Floriana) at doses of 12.5 μg/mL, 25 μg/mL, and 50 μg/mL of MGP or cocoa powder (control).

**Figure 5 foods-12-03153-f005:**
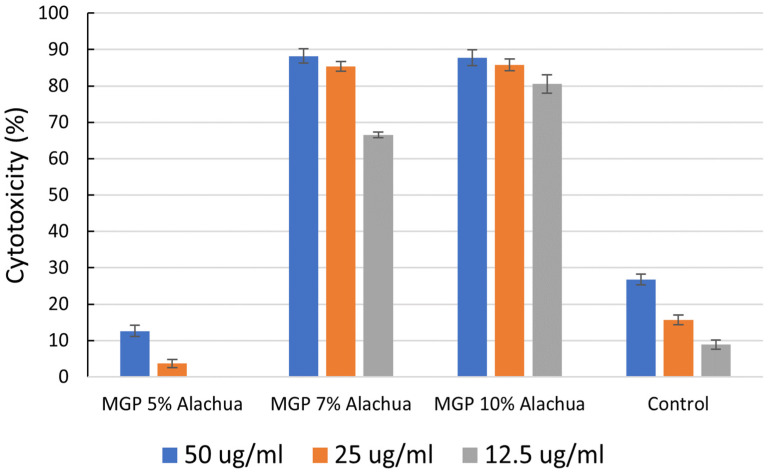
Inhibition percentage of triple-negative breast cancer cells MDA-MB-468 (MM-468) (African American) at doses of 12.5 μg/mL, 25 μg/mL, and 50 μg/mL of Alachua MGP chocolates or control chocolate sample.

**Table 1 foods-12-03153-t001:** The total phenolic content (TPC) and total flavonoid content (TFC) in MGP chocolates with varied concentrations of three varieties of unmacerated MGP dry powder and one macerated MGP dry powder. Different letters indicate significant difference at *p*-value < 0.05.

MGP Variety	MGP Concentration (*w*/*w*)	Macerated (Y/N)	TPC (mg GAE/g FW)	TFC (mg GAE/g FW)
Senoria	5%	N	0.33 ± 0.06 ^d^	0.13 ± 0.00 ^b^
Senoria	7%	N	0.21 ± 0.02 ^b^	0.13 ± 0.00 ^b^
Senoria	10%	N	0.21 ± 0.08 ^bc^	0.13 ± 0.00 ^b^
Floriana	5%	Y	0.28 ± 0.04 ^c^	0.16 ± 0.01 ^c^
Floriana	7%	Y	0.48 ± 0.04 ^e^	0.18 ± 0.00 ^d^
Floriana	10%	Y	0.52 ± 0.01 ^f^	0.20 ± 0.00 ^e^
Floriana	5%	N	0.13 ± 0.02 ^b^	0.13 ± 0.01 ^b^
Floriana	7%	N	0.32 ± 0.11 ^d^	0.16 ± 0.00 ^c^
Floriana	10%	N	0.04 ± 0.02 ^a^	0.12 ± 0.00 ^b^
Alachua	5%	N	0.15 ± 0.04 ^bc^	0.12 ± 0.00 ^b^
Alachua	7%	N	0.23 ± 0.14 ^bc^	0.14 ± 0.03 ^b^
Alachua	10%	N	0.02 ± 0.01 ^a^	0.10 ± 0.00 ^a^
Control	0%	N	0.36 ± 0.04 ^d^	0.13 ± 0.00 ^b^

**Table 2 foods-12-03153-t002:** The antioxidant activities (measured by DPPH and FRAP approaches) of MGP chocolates with varied concentrations of three varieties of unmacerated MGP dry powder and one macerated MGP dry powder. Different letters indicate significant difference at *p*-value < 0.05.

MGP Variety	MGP Concentration (*w*/*w*)	Macerated (Y/N)	DPPH %	FRAP (mM TE/g FW)
Senoria	5%	N	36.18 ± 3.66 ^abc^	316.13 ± 2.80 ^b^
Senoria	7%	N	31.98 ± 3.55 ^ab^	308.80 ± 8.07 ^ab^
Senoria	10%	N	43.49 ± 3.05 ^cd^	341.87 ± 7.77 ^cd^
Floriana	5%	Y	36.45 ± 6.31 ^abc^	360.00 ± 3.49 ^e^
Floriana	7%	Y	41.95 ± 4.20 ^c^	338.53 ± 6.55 ^cd^
Floriana	10%	Y	50.72 ± 6.32 ^d^	404.87 ± 14.12 ^g^
Floriana	5%	N	26.55 ± 4.03 ^a^	338.40 ± 6.12 ^cd^
Floriana	7%	N	45.35 ± 8.33 ^c^	380.40 ± 5.44 ^f^
Floriana	10%	N	29.43 ± 6.58 ^ab^	304.33 ± 2.83 ^ab^
Alachua	5%	N	33.93 ± 8.09 ^abc^	335.20 ± 2.99 ^c^
Alachua	7%	N	21.67 ± 9.15 ^a^	346.53 ± 4.87 ^d^
Alachua	10%	N	21.27 ± 6.71 ^a^	287.40 ± 17.51 ^a^
Control	0%	N	43.41 ± 3.20 ^c^	328.73 ± 6.71 ^b^

## Data Availability

The data presented in this study are available on request from the corresponding author.

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
