# Peer review of "Investigation of Antioxidant and Cytotoxicity Activities of Chocolate Fortified with Muscadine Grape Pomace"

_foods, 2023, doi:10.3390/foods12173153_

Round 1
Reviewer 1 Report
This manuscript is about investigate antioxidant and anticancer activities of chocolate fortified with Muscadine grape pomace (MGP). It is interesting and I think this manuscript needs major revision. You can find my comments in below:
1. The manuscript must be revised grammatically and the English level of it must be improved by a native editor.
2. The authors must re-write the abstract and conclusion sections.
3. In line 51, add the mechanism of reducing fat content.
4. In lines 62 to 66, it is better to give more details from previous studies. The author only gives detail about one simple reference.
5. In line 76, what is the mechanism of effect of processing containing thermal or high shear on the antioxidant and anticancer activities.
6. In lines 117 to 128, please add more details about preparation of cocoa beans and chocolate extracts.
7. Why did the authors only evaluate antioxidant activity by DPPH and FRAP methods and others like ABTS were not used.
8. In results and discussion part (3.2 and 3.3 sections), please add more details and references. It is better to add mechanisms and reason of for example, increasing and decreasing antioxidant and anticancer related values.
9. Please increase the DPI values of figures. The quality of them is poor.
The manuscript must be revised grammatically and the English level of it must be improved by a native editor.
Author Response
1- The manuscript must be revised grammatically and the English level of it must be improved by a native editor.
We want to thank the reviewer for his suggestion. However, we revised the manuscript grammatically.
2-The authors must re-write the abstract and conclusion sections.
The authors have read the abstract and conclusion and we re-wrote it.
3- In line 51, add the mechanism of reducing fat content.
The mechanism has been added with reference.
4-In lines 62 to 66, it is better to give more details from previous studies. The author only gives detail about one simple reference.
More details from previous studies have been added.
5- In line 76, what is the mechanism of the effect of processing containing thermal or high shear on the antioxidant and anticancer activities.
We have added more information related to the effect of high thermal or high shear on antioxidant and anticancer activities.
6- In lines 117 to 128, please add more details about the preparation of cocoa beans and chocolate extracts.
More details have been added.
7- Why did the authors only evaluate antioxidant activity by DPPH and FRAP methods and others like ABTS were not used.
We appreciate the reviewer’s comment. However, these two different methods (DPPH and FRAP) generally target most of the antioxidant metabolites.
8- In results and discussion part (3.2 and 3.3 sections), please add more details and references. It is better to add mechanisms and reason of for example, increasing and decreasing antioxidant and anticancer related values.
More details and references have been added.
9- Please increase the DPI values of the figures. The quality of them is poor.
We have changed all the figures with significantly improved DPI values.
Reviewer 2 Report
1. Giving the section “Abbreviation” in the beginning of manuscript
2. Page 3, line 93: Giving the meaning of FAMU.
3. Page 3, line 109: at 20 ºC to solidify=> at 20 ºC to solidify
4. Page 5, line 199: The TPC And TFC….. => The TPC and TFC……
5. Page 8, line 274: …..at the dose of 12.5 ug/ml, 25 ug/ml, and 50 ug/ml=> …..at the dose of 12.5 ug/mL, 25 ug/mL, and 50 ug/mL. ml or mL? consistency through the manuscript.
6. Please clearly discuss the correlation or relationship between antioxidant activity based on DPPH and FRAP, anticancer activities of MGP chocolates and their total phenolic compounds (TPC) and total flavonoid content (TFC).
Author Response
- Giving the section “Abbreviation” in the beginning of manuscript
We have checked and defined the Abbreviation in the content of the manuscript since a separate Abbreviation section is not required by the journal.
- Page 3, line 93: Giving the meaning of FAMU.
We have added the meaning of FAMU as suggested.
- Page 3, line 109: at 20 ºC to solidify=> at 20 ºC to solidify
We have fixed the problem as suggested.
- Page 5, line 199: The TPC And TFC….. => The TPC and TFC……
We have fixed the problem as suggested.
- Page 8, line 274: …..at the dose of 12.5 ug/ml, 25 ug/ml, and 50 ug/ml=> …..at the dose of 12.5 ug/mL, 25 ug/mL, and 50 ug/mL. ml or mL? consistency through the manuscript.
We have fixed the consistency problem as suggested.
- Please clearly discuss the correlation or relationship between antioxidant activity based on DPPH and FRAP, anticancer activities of MGP chocolates and their total phenolic compounds (TPC) and total flavonoid content (TFC).
We have added more relevant discussion on the connections between TPC, TFC, and anticancer activities as suggested.
Reviewer 3 Report
The MS is simple; however, the industrial application. MS required a thorough revision. Address the following points:
1. Change the title: The title is not appropriate. Representation of anticancer activity by merely cell cytotoxicity assessment is not appropriate.
2. To make the abstract more informative, provide some factual results such as TPC quantification, DPPH inhibition, and radical scavenging IC50 values. Also, add some concluding lines in the abstract section.
3. Rewrite the keywords, as all the keywords are merely a replication of the title. Avoid the keywords (TNBC, Anticancer, Antioxidant, Pomace) that are already mentioned in the title.
4. The introduction is too lengthy and contains some irrelevant text(such as lines 53-66). It is better to shorten the introduction by deleting the relevant section or transfer it to the discussion section.
5. For methodology section make schematic diagram.
6. Use the same format to represent mathematical units such as 12 % (line 49) or 96% line 143); µL or µl; ml or mL; address the issue throughout the MS.
7. In line no 188, explain how the IC50 value was calculated briefly.
8. In Figure 1, write the name of the bar column “Macerated” (in the TPC graph) as mentioned in the TFC graph.
9. For all the figures, make the figure captions more informative. Represent the graphs in Figures 1 and 2 as A and B.
10. It would be better if the author could provide the IC50 values for the DPPH antioxidant assay.
11. What are the criteria for selecting 5, 7, and 10% concentration, as depicted in Table 2? For Figures 3 and 4, explain how IC50 was calculated.
12. Rewrite the conclusion thoroughly in the context of the present findings.
Minor english editing
Author Response
1- Change the title: The title is not appropriate. Representation of anticancer activity by merely cell cytotoxicity assessment is not appropriate.
The title has been changed to “Investigation of Antioxidant and Cytotoxicity Activities of Chocolate Fortified with Muscadine Grape Pomace (MGP)”.
2-To make the abstract more informative, provide some factual results such as TPC quantification, DPPH inhibition, and radical scavenging IC50 values. Also, add some concluding lines in the abstract section.
All required results have been added to make the abstract more informative.
3- Rewrite the keywords, as all the keywords are merely a replication of the title. Avoid the keywords (TNBC, Anticancer, Antioxidant, Pomace) that are already mentioned in the title.
The keywords have been modified as per the reviewer’s suggestion.
4- The introduction is too lengthy and contains some irrelevant text (such as lines 53-66). It is better to shorten the introduction by deleting the relevant section or transfer it to the discussion section.
The introduction has been improved.
5- For the methodology section make a schematic diagram.
We have added a schematic diagram for the methodology as suggested.
6- Use the same format to represent mathematical units such as 12 % (line 49) or 96% line 143); µL or µl; ml or mL; address the issue throughout the MS.
All the required modifications have been done as per the reviewer’s comment.
7- In line no 188, explain how the IC50 value was calculated briefly.
The IC50 has been calculated using the software AAT Bioquest (https://www.aatbio.com/tools/ic50-calculator).
8- In Figure 1, write the name of the bar column “Macerated” (in the TPC graph) as mentioned in the TFC graph.
We have fixed the problem as indicated.
9- For all the figures, make the figure captions more informative. Represent the graphs in Figures 1 and 2 as A and B.
We have added more information to the figure captions to make them more informative.
10- It would be better if the author could provide the IC50 values for the DPPH antioxidant assay.
The authors declare that we used only one concentration due to the lack of the amount of the yield extract.
11- What are the criteria for selecting 5, 7, and 10% concentration, as depicted in Table 2? For Figures 3 and 4, explain how IC50 was calculated.
The percentages of MGP were chosen based on our preliminary studies, which indicated that at these contractions, the color and sensory attributes (sourness and overall likeness) of the MGP chocolates were not significantly different from the control. We have added some explanations in the manuscripts. The IC50 has been calculated using the software AAT Bioquest (https://www.aatbio.com/tools/ic50-calculator). We have added that information in the manuscript.
- Rewrite the conclusion thoroughly in the context of the present findings.
The authors have read the conclusion and we re-wrote it.
Reviewer 4 Report
Broad comments:
The article (foods-2487394) entitled “Investigate the Antioxidant and Anticancer Activities of Chocolate Fortified with Muscadine Grape Pomace (MGP)” aims to investigate the antioxidant/anticancer activity of muscadine grape pomace mixed with chocolate and the role of their phenolic and flavonoid contents in applying these properties in triple-negative breast cancer (TNBC) cells. Muscadine grape pomace and mixed products with chocolate extracts from three muscadine genotypes were investigated for total phenolic content (TPC), total flavonoid content (TFC), antioxidant capacity, and anticancer effects using African Americans breast cancer cells. The antioxidant activity was associated with high TPC content. The anticancer and antioxidant effects of muscadine grape pomace and chocolate are attributed to the TPC of extracts, which showed a stronger positive correlation with growth inhibition of African American breast cancer cells. However, I think that the results need to be better presented.
I suggest to change the title to:
Investigation of the Antioxidant and Anticancer Activities of Chocolate Fortified with Muscadine Grape Pomace
Figure and table captions:
Captions should contain a description allowing a reader to understand the figure or table. Any figure or table is considered incomplete without a proper caption.
Please add proper captions.
Please give details about using different letters and P values in figures and tables.
The article is commonly well written.
Author Response
1- I suggest changing the title to Investigation of the Antioxidant and Anticancer Activities of Chocolate Fortified with Muscadine Grape Pomace.
The title has been changed to “Investigation of Antioxidant and Cytotoxicity Activities of Chocolate Fortified with Muscadine Grape Pomace (MGP)”.
2- Captions should contain a description allowing a reader to understand the figure or table. Any figure or table is considered incomplete without a proper caption.
We have added more information to the figure and table captions to make them more informative.
3- Please give details about using different letters and P values in figures and tables.
We have added that information in the captions of the figures and tables. In addition, we have given the significant level for p values.
Round 2
Reviewer 3 Report
The MS has been substantially improved based on the raised comments. I recommend for the publication of the article.